

# Epoch-by-epoch estimation and analysis of BDS receiver differential code biases with the additional BDS-3 observations

Qisheng Wang[1,2,3], Shuanggen Jin[2,3,4*], Youjian Hu[1]

1. School of Geography and Information Engineering, China University of Geosciences, Wuhan 430074, China

2. School of Remote Sensing and Geomatics Engineering, Nanjing University of Information Science and Technology, Nanjing 210044, China

3. Jiangsu Engineering Center for Collaborative Navigation/Positioning and Smart Applications, Nanjing 210044, China

4. Shanghai Astronomical Observatory, Chinese Academy of Sciences, Shanghai 200030, China

*Correspondence to:* Shuanggen Jin (sgjin@nuist.edu.cn; sg.jin@yahoo.com) and Youjian Hu (hyj_06@163.com)

**Abstract.** The differential code bias (DCB) of global navigation satellite systems (GNSS) is an important error source in ionospheric modeling, which was generally estimated as constants every day. However, the receiver DCB may be changing due to the varying space environments and temperatures. In this paper, the receiver DCB of BeiDou Navigation Satellite System (BDS) is estimated as the changing parameter within one day with epoch-by-epoch. The BDS receiver DCBs are analyzed from 30 days of multi-GNSS experiment observations. The comparison of estimated receiver DCB of BDS with the DCB provided by German Aerospace Center (DLR) and Chinese Academy of Sciences (CAS) shows a good agreement. The root mean square (RMS) values of receiver DCB are 0.43 and 0.80ns with respect to DLR and CAS, respectively. In terms of the intra-day variability of receiver DCB, most of the receiver DCBs show relative stability within one day with the intra-day standard deviation (STD) of less than 1ns. However, larger fluctuations with more than 2ns of intra-day receiver DCB are found. Besides, the intra-day stability of receiver DCB calculated by the third-generation BDS (BDS-3) and the second-generation BDS (BDS-2) observations is compared. The result shows that the intra-day stability of BDS-3 receiver DCB is better than that of BDS-2 receiver DCB.

## 1. Introduction

The Global Positioning System (GPS) has been widely used as a useful tool in ionospheric monitoring and modeling (Hernández-Pajares et al., 1999;McCaffrey et al., 2017;Choi and Lee, 2018). With its rapid development in recent years, the BeiDou navigation satellite system (BDS) begins to play an important role in global navigation, positioning, timing and related applications (Su and Jin, 2019;Wang et al., 2019). Sine the third-generation BeiDou navigation satellite system (BDS-3) provided global services at the end of 2018, more and more stations have been constructed to track the BDS signals. In addition, BDS can also provide available observations globally for ionospheric modeling and research. The ionospheric total electron contents (TEC) can be obtained by the geometry-free linear combination of the dual-frequency global navigation satellite system (GNSS) code observations, in which the differential code bias (DCB) is one of the main error terms to be estimated and removed(Sanz et al., 2017).



In general, DCB is considered as an unknown parameter together with ionospheric parameters in global or region ionospheric modeling(Jin et al., 2012). However, only limited tracking stations can be used for early the regional system (BDS-2) because of its regional coverage(Montenbruck et al., 2013). In a method called IGG (Institute of Geodesy and Geophysics, Chinese Academy of Sciences), which was proposed to estimate the BDS DCB, the single-station ionosphere
modeling method is used to overcome a limited number of global tracking stations for BDS(Li et al., 2012). Furthermore, in order to make full use of the useful information of the raw measurement, a method based on triple-frequency uncombined precise point positioning(UPPP) is used to estimate the BDS DCB(Shi et al., 2015;Fan et al., 2017;Liu et al., 2019). All the above methods need to estimate the ionospheric TEC simultaneously in DCB estimation. Therefore, an existing global ionospheric map (GIM) is used to eliminate ionospheric TEC for improving the BDS DCB estimation efficiency (Jin et al.,
2016;Li et al., 2019;Wang et al., 2019). At present, two International GNSS Service (IGS) analysis centers can provide DCB products of BDS. One is the German Aerospace Center (DLR), in which the DCB products are generated by the GIM based on the Multi-GNSS Experiment (MGEX) observational data(Montenbruck et al., 2014). The other IGS analysis center is the Chinese Academy of Sciences (CAS), where the DCB products are estimated by the IGG method based on the multi-GNSS observations(Wang et al., 2015). The BDS DCB products provided by DLR and CAS can be obtained at
ftp://cddis.gsfc.nasa.gov/pub/gps/products/mgex/dcb/. It should be noted that some of the MGEX stations are selected for the DCB estimation of DLR and CAS; but the number of selected stations is different for DLR and CAS. In addition, only the receiver DCB of stations used in DLR and CAS can be provided in their DCB products.

In the DCB estimation, the satellite and receiver differential code biases (DCB) of BDS are generally estimated as constants every day(Li et al., 2014;Xue et al., 2015). However, the receiver DCB may be various within one day due to
varying space environments and temperatures. As can be seen from previous studies, the stability of BDS receiver DCB is not as good as that of the satellite(Jin et al., 2016;Xue et al., 2015). Thus, a study on estimation and analysis of short-term variations of multi-GNSS DCB was carried out by Li(Li et al., 2018), who used the GIM of IGS and the satellite DCB of DLR to estimate the receiver DCB with an hourly resolution. As shown by the results, an obvious short-term variation was found in the receiver DCB within one day. However, the receiver DCB may exhibit variations within one hour. Furthermore,
more global available observations from BDS-3 can be used in DCB estimation, instead of using the only BDS-2 in previous studies. Since 2013, the BDS measurements have been collected by the MGEX network(Montenbruck et al., 2017). Currently, BDS measurements can be tracked by more and more stations of MGEX benefited by the rapid expansion of BDS-3.

In order to better analyze the short-term variations of BDS receiver DCB with the additional BDS-3 observations, this
paper presents a DCB estimation method based on the GIM to estimate the satellite and receiver DCB. The receiver DCB that is treated as the changing parameter within one day is estimated epoch-by-epoch. The intra-day stability of receiver DCBs are analyzed through BDS-2+BDS-3 observations. Besides, the intra-day stability of the receiver DCB obtained by BDS-2-only and BDS-3-only observations are compared. This rest of the paper is organized as follows. In Section 2, the



DCB estimation method and the experiment are introduced. In Section 3, the experiment results and corresponding analysis
are shown. Finally, conclusions are given in Section 4.

## 2. Method and Data

In this section, the method used to estimate receiver DCB epoch-by-epoch is firstly introduced. The corresponding equations
for the DCB estimation method are described in detail. Then, the experimental data used in this study are described.

### 2.1 DCB estimation method.

As we all know, the ionospheric observation equation can be obtained by the geometry-free linear combination of dual-
frequency GNSS code observations. The code observations smoothed by carrier phase can be expressed as
follows(Wellenhof et al., 1992;Hernández-Pajares et al., 2017;Hernández-Pajares et al., 2009):

$$\tilde{P}_{4,r}^s(t) = \tilde{P}_{2,r}^s(t) - \tilde{P}_{1,r}^s(t) = 40.28(\frac{1}{f_1^2} - \frac{1}{f_2^2}) \cdot STEC_r^s(t) + c \cdot (DCB_r + DCB^s) \tag{1}$$

where $s$, $r$ and $t$ denote the satellite, receiver and epoch, respectively; $f_1$ and $f_2$ are the first and second frequency of the
observations, respectively; $\tilde{P}_{1,r}^s$ and $\tilde{P}_{2,r}^s$ are smoothed code observations at frequency $f_1$ and $f_2$, respectively; $\tilde{P}_{4,r}^s$ is the

smoothed geometry-free linear combination of code observations; $c$ is the speed of light; $DCB_r$ and $DCB^s$ are the receiver

and satellite DCB, respectively; $STEC_r^s$ is the slant total electron content along the signal propagation path between GNSS

satellite $s$ and ground receiver $r$; The slant total electron content can be converted to the vertical TEC by a mapping

function, which can be defined as follows(Schaer, 1999):

$$\begin{cases} STEC = VTEC \cdot M(z) \\ M(z) = \frac{1}{\cos z'}, \sin z' = \frac{R}{R+H}\sin z \end{cases} \tag{2}$$

where $R$ is the mean radius of the Earth; $H$ is the height of assumed single-layer ionosphere;

$z$ and $z'$ denote the satellite zenith angles of a satellite at the receiver and the corresponding the ionospheric pierce point
(IPP), respectively.

According to Eq (1), the satellite and receiver DCBs can be obtained by estimation together with the ionospheric
parameters in ionospheric modeling or by elimination of the STEC with the existing GIM. The satellite and receiver DCBs
are generally estimated as constants every day. In order to consider the intra-day variability of the receiver DCB, the receiver
DCB that is treated as the changing parameter within one day is estimated epoch-by-epoch in this paper. And the satellite
DCB is still estimated as a constant over one day, which can be defined as follows:

$$DCB_r(t) + DCB^s = (\tilde{P}_{4,r}^s(t) - 40.28(\frac{1}{f_1^2} - \frac{1}{f_2^2}) \cdot VTEC_r^s(t) \cdot M(z)) / c \tag{3}$$



where *VTEC* values can be eliminated with the GIM provided by IGS. Therefore, Eq (1) can be further described as follows:

$$
\begin{bmatrix}
IR_1 & \cdots & \cdots & IS_1 \\
\vdots & \ddots & \vdots & \vdots \\
\cdots & \cdots & IR_r & IS_r
\end{bmatrix}
\times
\begin{bmatrix}
DCB_1(t1) \\
\vdots \\
DCB_r(tr) \\
DCB^1 \\
\vdots \\
DCB^s
\end{bmatrix}
=
\begin{bmatrix}
(\tilde{P}^s_{4,r}(1) - k \cdot VTEC^s_r(1))/c \\
\vdots \\
(\tilde{P}^s_{4,r}(m) - k \cdot VTEC^s_r(m))/c
\end{bmatrix}
\tag{4}
$$

where $IR_r$ and $IS_r$ are the coefficient matrix of receiver DCB and satellite DCB for the r-th station, respectively, which

consist of 0 and 1; The sizes of $IR_r$ and $IS_r$ are $mr \times tr$ and $mr \times s$ ,respectively; $mr$ is the number of available observations

for the r-th station, $m = m1 + \cdots mr$ is the total of available observations for all stations and $s$ is the total number of all

available satellites; $tr$ is the number of available epoch; $DCB_r(tr)$ is the vector of $1 \times tr$ ; $DCB^s$ is the DCB of the s-th

satellite; $k = 40.28(\dfrac{1}{f_1^2} - \dfrac{1}{f_2^2}) \cdot M(z)$ . As can be seen from Eq (4), the number of parameters to be estimated is $t1 + \cdots tr + s$ ,

and $mr > tr$ , $m > t1 + \cdots tr + s$ . The number of observations is much larger than the number of parameters to be estimated.

Therefore, the receiver and satellite DCB values can be obtained based on the least square method. Notably, the zero-mean

condition should be added to the satellite DCB in least square adjustment due to the correlation between satellite and

receiver DCB, which can be defined as below:

$$
\sum_{i=1}^{s} DCB^i = 0
\tag{5}
$$

In order to reduce the influence of multipath noise and mapping function error, we set a 20°cutoff elevation angle. In

addition, the weight of observations is used in DCB estimation, which can be expressed as follows:

$$
P = \frac{1}{(1 + \cos^2 E)}
\tag{6}
$$

where $P$ is the weight of DCB observations and $E$ is elevation angle of the satellite.

     Based on Eqs (4)-(6), the DCB values of the epoch-by-epoch receiver and daily satellite can be obtained. In this study,

the BDS receiver and satellite DCBs with the additional BDS-3 observations are estimated by the method mentioned above.

The intra-day stability of receiver DCBs are analyzed by using BDS-2+BDS-3 observations. Besides, the intra-day stability

of receiver DCB obtained by using BDS-2-only and BDS-3-only observations are compared.

## 2.2 Experimental Data

In order to verify the performance of DCB estimation method and analyze the intra-day stability of BDS receiver with the

additional BDS-3 observations, we use 30 days (January 2019) of multi-GNSS experiment observations, which include 109





stations. It should be noted that the DCB estimation and analysis in this study are mainly for C2I-C6I DCB since the C2I and C6I code observations from BDS-2 and BDS-3 can be tracked simultaneously by more stations. Figure 1 presents the distribution of selected MGEX stations in January 2019, in which different types of receivers are distinguished by colors. And the corresponding information of receiver is listed in Table 1. As can be seen from Figure 1 and Table 1, the TRIMBLE receiver is used in most stations. In addition, the average number of available observation epoch for the selected MGEX stations in January 2019 is counted, as shown in Figure 2. It is clear that greater quantities of available observations are

shown in the stations located in the Asia Pacific region.

## 3. Results and Analysis

The section begins by verifying the performance of the DCB estimation method, in which the satellite and receiver DCBs are compared with the DCB products of CAS and DLR. Then, the intra-day stability of the receiver DCB is analyzed. The intra-day stability of receiver DCB obtained by using BDS-2-only and BDS-3-only observations are finally compared.

### 3.1 Validation of DCB estimation method.

To verify the performance of the DCB estimation method, we take the DCB products from CAS and DLR as references, and use the mean difference and RMS values to evaluate the estimated satellite and receiver DCBs. Figure 3 shows the mean difference and RMS of the estimated satellite DCB with respect to CAS and DLR. It can be seen that the estimated satellite DCB shows a good agreement with CAS and DLR. The mean differences are mostly within ±0.2ns, and their mean values with respect to DLR and CAS are -0.004 and -0.006ns, respectively. The RMS values are less than 0.4ns, and the mean RMS

values with respect to DLR and CAS are 0.18 and 0.23ns, respectively. It can be concluded that the DCB estimation method can get the satellite DCB that is in good agreement with DLR and CAS. In other words, it has little influence on the estimation result of satellite DCB when the receiver DCB is estimated epoch-by-epoch.

Figure 4 and Figure 5 present the mean difference and RMS of the estimated BDS receiver DCB with respect to CAS and

DLR. It should be noted that the receive DCB for some of our selected MGEX stations cannot be provided by CAS and DLR since these stations were not used in their DCB estimation. In this study, 101 stations' reference values can be provided by CAS and 61 stations' reference values can be provided by DLR. As it can be seen, when DLR is taken as a reference, the mean difference mostly ranges from -0.5ns to 0.5ns, and their mean value is 0.05ns. The corresponding RMS values are mostly less than 0.5ns and the mean RMS is 0.43ns. However, the difference between our estimated receiver DCB and CAS shows a larger value. The difference ranges from -2ns to 2ns, the values for some stations located in low latitude are more

than 1ns. In terms of RMS, the values for most stations are less than 1ns, and the corresponding mean RMS is 0.80ns. The reason may be related to the single-station ionosphere modeling adopted by CAS, since the modeling accuracy of stations in low latitude is low. The evaluation results show that the DCB estimation method can obtain the receiver DCB value with certain accuracy.





## 3.2 Intra-day stability of receiver DCB.

In this study, the time series values of the receiver DCB within one day can be obtained through the epoch-by-epoch estimation for the receiver DCB. In order to better understand the intra-day variations and stability of receiver DCB, we calculate and analyze the intra-day standard deviation (iSTD) and the intra-day fluctuation need as:

$$iSTD = \sqrt{\frac{\sum\left(RDCB_i - \overline{RDCB}\right)}{N-1}}$$

$$fDCB_i = RDCB_i - RDCB_1$$

(7)

where $iSTD$ is the intra-day STD, $RDCB_i$ is the receiver DCB at epoch i, $\overline{RDCB}$ is the corresponding mean value of one day, $N$ is the number of the available epoch, $fDCB_i$ is the corresponding value of fluctuation at epoch i and $RDCB_1$ is the receiver DCB at the first available epoch.

Four consecutive days (January 2, 3, 4 and 5 in 2019) are taken as an example. The intra-day STD of receiver DCB for all stations on the four days is shown in Figure 6 and the statistical results of intra-day STD are listed in Table 2. As can be seen from Figure 6, the intra-day STD of receiver DCB for all stations shows similar values on four consecutive days. This result can also be shown in Table 2, with similar minimum, maximum and mean values of intra-day STD on the four days. It indicates that the intra-day stability of the receiver DCB for most stations is almost the same on the four days.

As can also be seen from Figure 6, the intra-day STD of the receiver DCB for most stations is less than 1ns on the four consecutive days. In particular, the intra-day STD of the receiver DCB for the stations located in the Asia Pacific region are significantly less than that for other regions. The reason may be related to greater quantities of available observations of these stations in the Asia Pacific region. In addition, some stations with larger intra-day STD of the receiver DCB can be found in low-latitude regions, which may be related to the fact that the ionosphere is more active at low latitude(Li et al., 2018). It can be concluded that the available observations of the station and the level of ionospheric activity of the area where the station is located may be two important factors affecting the intra-day stability of the receiver DCB.

Figure 7 shows the intra-day fluctuation and intra-day STD of the receiver DCB for the selected 9 stations located in the Asia Pacific region within 30 days. It can be seen that the intra-day fluctuation of the receiver DCB is mostly within ±1ns, and the corresponding intra-day STD is mostly less than 0.4ns. It is clear that the receiver DCB with larger fluctuation shows lower intra-day stability and vice versa. Although intra-day receiver DCB of the 9 selected stations are relatively stable, larger fluctuations with more than 2ns of intra-day receiver DCB can be found.

In order to better explain the intra-day stability of the receiver DCB, Figure 8 shows the statistical distribution of intra-day STD of the receiver DCB for all stations on 30 days, where |0.5|, |1| and |2| represent the percentage of the absolute value of intra-day STD is less than 0.5, 1, and 2ns, respectively. More than 94% of the intra-day STDs are less than 1ns, indicating that most of the receiver DCBs show relative stability within one day with the intra-day STD of less than 1ns.





### 3.3 Comparison between BDS-2-only and BDS-3-only solutions.

As we all know, the BDS-3 is a new generation satellite navigation system. The additional BDS-3 observations used in this study contribute to the increasing available observables in DCB estimation. To further understand the contribution of BDS-3, we compare the intra-day stability of the receiver DCB obtained by using BDS-2-only and BDS-3-only observations. Figure 9 and Figure 10 show the mean intra-day STD of receiver DCB and the corresponding Statistical distribution for all stations within 30 days by using BDS-2-only and BDS-3-only observations. Obviously, the result shows that the intra-day stability of BDS-3 receiver DCB is better than that of BDS-2 receiver DCB.

### 4. Conclusions.

The short-term variations of epoch-by-epoch BDS receiver DCB are estimated and analyzed with the additional BDS-3 observations. The BDS receiver DCBs with the additional BDS-3 observations are analyzed through 30 days of multi-GNSS experiment observations. To verify the performance of the DCB estimation method, we take the DCB products from CAS and DLR as references, and use the mean difference and RMS values to evaluate the estimated satellite and receiver DCBs. The estimated satellite DCB shows a good agreement with CAS and DLR, since the mean differences are mostly within ±0.2ns, and their mean values with respect to DLR and CAS are -0.004 and -0.006ns, respectively. The RMS values are less than 0.4ns, and the mean RMS values with respect to DLR and CAS are 0.18 and 0.23ns, respectively. In terms of the intra-day variability of the receiver DCB, most of the receiver DCBs show relative stability within one day with the intra-day STD of less than 1ns. However, larger fluctuations with more than 2ns of intra-day receiver DCB can be found. It can be concluded that the available observations of the station and the level of ionospheric activity of the area where the station is located may be two important factors affecting the intra-day stability of the receiver DCB. As shown by the results of the intra-day stability of receiver DCB obtained by using BDS-2-only and BDS-3-only observations, the intra-day stability of BDS-3 receiver DCB is better than that of BDS-2 receiver DCB.

*Data availability.* The MGEX station data can be downloaded from ftp://cddis.gsfc.nasa.gov/pub/gps/data/, last access: 20 May 2020.

*Author contributions.* Qisheng Wang carried out the analysis and wrote the paper, All the co-authors helped in the interpretation of the results, read the paper, and commented on it.

*Competing interests.* The authors declare that they have no conflict of interest.

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

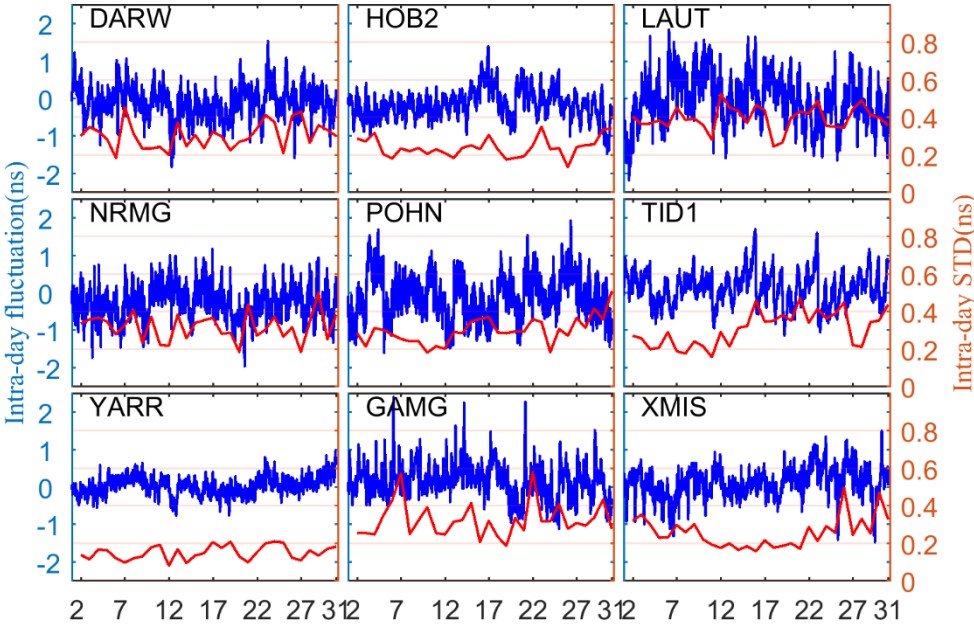

**Figure. 7 The intra-day fluctuation (blue line) and intra-day STD (red line) of the receiver DCB for the selected 9 stations within 30 days**

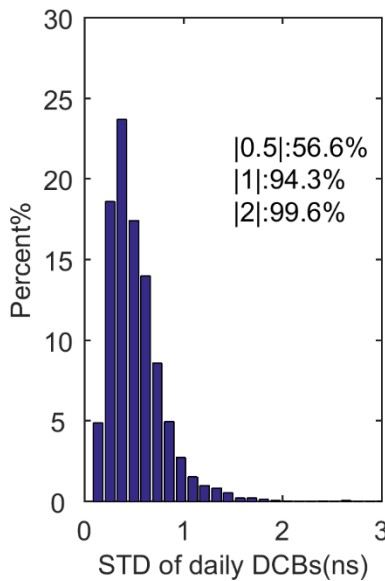

**Fig. 8 Statistical distribution of intra-day STD of receiver DCB for all station in 30 days**





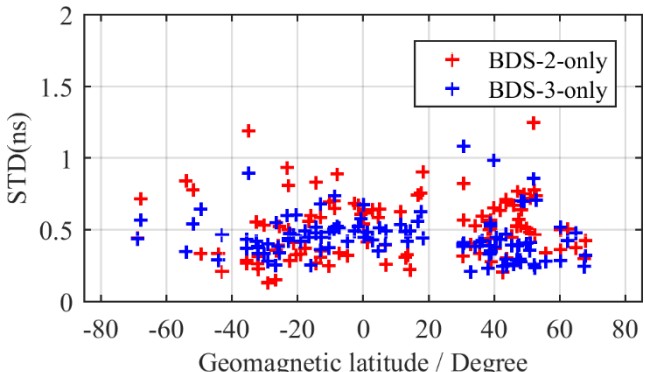

**Figure.9 The mean intra-day STD of receiver DCB for all station in 30 days by using BDS-2-only and BDS-3-only observations**

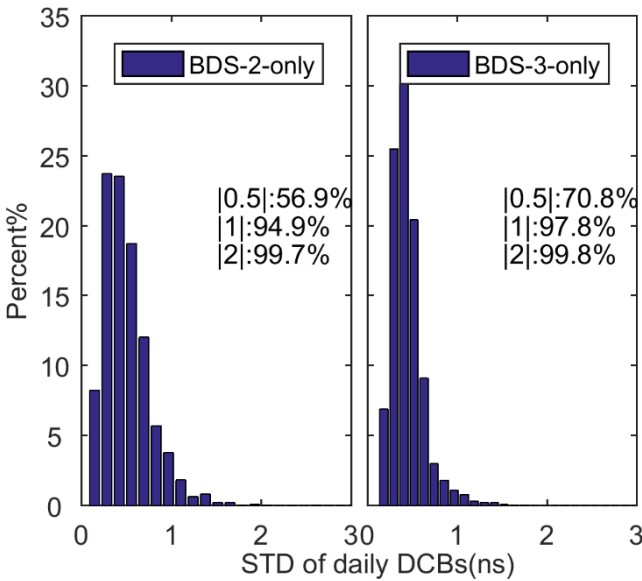

**Figure.10 Statistical distribution of intra-day STD of receiver DCB by using BDS-2-only and BDS-3-only observations**

280

**Table 1 Summary of selected MGEX stations in January 2019**

| Index | Receiver type | Number of stations |
|---|---|---|
| 1 | JAVAD TRE_3 | 5 |
| 2 | JAVAD TRE_3 DELTA | 17 |
| 3 | SEPT POLARX5 | 31 |
| 4 | SEPT POLARX5TR | 14 |



| 5 | TRIMBLE ALLOY | 2 |
| 6 | TRIMBLE NETR9 | 40 |

**Table 2 Statistical of intra-day STD of receiver DCB for all stations on four consecutive days**

| Date | January 2,2019 | January 3,2019 | January 4,2019 | January 5,2019 |
|---|---|---|---|---|
| Min iSTD | 0.14 | 0.11 | 0.14 | 0.16 |
| Max iSTD | 1.77 | 1.73 | 1.66 | 1.60 |
| Mean iSTD | 0.51 | 0.54 | 0.55 | 0.57 |

285