# Peer review of "Epoch-by-epoch estimation and analysis of BDS receiver differential code biases with the additional BDS-3 observations"

_Annales Geophysicae, 2020_

## Referee Comment (RC1) · Anonymous Referee #1 · 28 Jul 2020

This paper reports estimation of the receiver DCB of BeiDou Navigation Satellite System (BDS) as a changing parameter within the day with epoch-by-epoch. The authors have compared DCBs estimated with those provided by German Aerospace Center (DLR) and Chinese Academy of Sciences (CAS). Finally, the difference between the intra-day stability of receiver DCB estimated using (BDS-3) and (BDS-2) generations were calculated.

comments as follow:

- Put a space between the number and "ns" in the whole paper, like in line 15, "0.80 ns".

- Replace "sine" by since in line 25.

- It should be a space before the parentheses in the whole paper, like in line 30, "…..removed (Sanz et al., 2017)".

- The used calculation software or the programming language environment are not mentioned in the paper.

- What is the used value of the height of the single layer "$H$" in equation (2)?

- The data availability link of the satellite data is not exist in the paper, which is required to calculate the elevation angle of the satellite used in the weight function (equation 6).

- In the post processing programs, one value of the satellite and receiver DCB is used (the mean through the day), so the authors should clearly show the importance and the applications of the epoch by epoch DCB values.

- It should provide the used GIM data source link.

- It should mentioned in the abstract and the conclusions that the calculations are based on the GIM of the IGS, because it is an important factor can affect the resulted DCB, which can be changed when using another GIM from other sources like CODE or JPL.

- The temporal resolution of the Ionex file is 1 hr (IGS) or 2 hrs (CODE) and the observation epoch is 30 sec, that means the ionosphere value still constant through number of calculated DCB, so did you try to calculate DCB from 1 hr file and 2 hrs file and compare the results.

---

## Referee Comment (RC2) · Anonymous Referee #2 · 11 Sep 2020

This paper presents a technique for estimating satellite and receiver DCBs on an epoch by epoch basis utilizing the BDS observation data and GIMs from DLR and CAS. A comparison is made between the estimated DCBs and the ones computed by DLR/CAS. Intra day stability and intra day fluctuation are presented for the estimated DCBs using both BDS-II and BDS III data.

Line 11: "the receiver DCB of BeiDou Navigation Satellite ...." This needs to be re-phrased to highlight the receiver DCBs are being estimated using the BeiDou system.

Line 23: Include a space between each in-text reference after ";". Do it for all the future instances.

Line 30: Put a space between an in-text reference and the other text.

Line 31: Replace "region" with "regional".

Line 33: As per your stated reference, it's called "IGGDCB" where IGG stands for Institute of Geodesy and Geophysics. Please review.

Line 45-46: "It should be noted that some of the ....." Sentence needs to be rephrased to make it clearer.

Line 49: "may be various" needs to be replaced with "may varies"

Line 52: "by Li et al. (2018)" instead of "Li (Li et al., 2018)."

Line 57: Point not clear. Please elaborate.

Line 60: "The rest of the paper ..." instead of "This rest of the paper ..."

Line 65: "are presented" instead of " are shown."

Line 78: Put a full stop after receiver r i.e. "r. The slant total electron content can be converted to ..."

Line 81: Put a numerical value against "R" and "H". Refer to Schaer (1999).

Line 82: Get rid of "the" after "corresponding."

Line 94: Put the equation like this: m = m1 + ..... + mr. It would have been better to use the subscripts here like  $m_1$ ,  $m_2$ ,  $m_r$ .

Line 102: Put a space between 20° and cutoff.

Line 106-109: How has this been carried out? Existing software or new software written. Please add some information about it.

Line 117: "And the corresponding ...." Don't start a sentence with the conjunction 'and'. Remove all future instances.

Line 118: "epochs" instead of "epoch."

Line 127: What is the period/duration over which this mean is computed? It must be stated somewhere in the text.

Line 135: It should be "receiver DCB" instead of "receive DCB".

Figure 4 and Figure 5: Put labels to differentiate between CAS and DLR based estimation.

Figure 7: Have you tried comparing the trend of the estimated DCBs with the GIM you have used? After all, GIMs have certain accuracy. The ionosphere follows this pattern of being least active during the night to highly active during the day. So, this could influence the estimation process and one might be able to pick this trend by comparing the plotting the estimated DCBs against the ionospheric TEC derived from the GIM.

Comment# My experience of working with DCBs is that these are quite stable over time and especially in a controlled environment. One can see big fluctuations if there is some hardware problem in the receiver circuitry. In my opinion, most of these MGEX stations are in a relatively controlled environment, so the chances of all the receivers behaving in the same manner are quite low. It would be good to explore this area further.

Please try plotting and comparing the DCBs for the stations which are utilising the same receiver type. Sept PolaRx5 could be good option with data available from 31 stations. I expect to see close results for these stations, no identical.

---

## Author Comment (AC1) · 17 Sep 2020

Dear Reviewer:

Thanks very much for your comments. These comments were all valuable and very helpful for revising and improving our paper. In the revised manuscript, we have carefully revised it. The following is a point-to-point response to the comments

Thank you very much!

•Put a space between the number and "ns" in the whole paper, like in line 15, "0.80 ns".

**Response:**

We have made corrections in the revised manuscript.

• Replace "sine" by since in line 25.

**Response:**

We have revised it in the revised manuscript.

• It should be a space before the parentheses in the whole paper, like in line 30, "…..removed (Sanz et al., 2017)".

**Response:**

We have made corrections in the revised manuscript.

• The used calculation software or the programming language environment are not mentioned in the paper.

**Response:**

The MATLAB processing program is used in the paper, which is developed based on the M_DCB software (Jin et al., 2012).
We have added it in the revised manuscript.

• What is the used value of the height of the single layer "H" in equation (2)?

**Response:**

The value of H is 450km. We have added it in the revised manuscript.

• The data availability link of the satellite data is not exist in the paper, which is required to calculate the elevation angle of the satellite used in the weight function (equation 6).

**Response:**

The precise satellite ephemeris is provided by Wuhan University, which is available at ftp://cddis.gsfc.nasa.gov/pub/gps/products/mgex/.
We have added the data source link in the revised manuscript.

• In the post processing programs, one value of the satellite and receiver DCB is used (the mean through the day), so the authors should clearly show the importance and the applications of the epoch by epoch DCB values.

**Response:**

In the DCB estimation, the satellite and receiver DCBs of BDS are generally estimated as constants every day. However, the receiver DCB may varies within one day due to varying space environments and temperatures. The estimation of receiver DCB as constant every day may cause errors in ionospheric modelling, if the receiver DCB has significant intra-day fluctuations. It would have been better to analyze the intra-day variation of receiver DCB before the estimation of receiver DCB as constant over a day. Thus, the intra-day variation analysis of BDS receiver DCB with the additional BDS-3 observations is carried out in the study.
We have added it in the revised manuscript.

• It should provide the used GIM data source link.

**Response:**

The GIM used is downloaded at ftp://cddis.gsfc.nasa.gov/pub/gps/products/ionex/.
We have added the data source link in the revised manuscript.

• It should mentioned in the abstract and the conclusions that the calculations are based on the GIM of the IGS, because it is an important factor can affect the resulted DCB, which can be changed when using another GIM from other sources like CODE or JPL.

**Response:**

Yes, the GIM is an important factor in our DCB estimation.
We have added it in the abstract and the conclusions of the revised manuscript.

• The temporal resolution of the Ionex file is 1 hr (IGS) or 2 hrs (CODE) and the observation epoch is 30 sec, that means the ionosphere value still constant through number of calculated DCB, so did you try to calculate DCB from 1 hr file and 2 hrs file and compare the results.

**Response:**

Yes, the temporal resolution of GIM is 2h or 1h. The GIM used in this study is from IGS's CODE with 1 h resolution. Since the GIM of CODE has higher temporal resolution, we did not calculate DCB using GIM with 2 h resolution.

---

## Author Comment (AC2) · 17 Sep 2020

Dear Reviewer:

Thanks very much for your comments. These comments were all valuable and very helpful for revising and improving our paper. In the revised manuscript, we have carefully revised it. The following is a point-to-point response to the comments

Thank you very much!

Line 11: "the receiver DCB of BeiDou Navigation Satellite …." This needs to be re-phrased to highlight the receiver DCBs are being estimated using the BeiDou system.

**Response:**

We have revised it as following:

"a method based on the GIM of IGS is presented to estimate the BDS receiver DCB with epoch-by-epoch"

Line 23: Include a space between each in-text reference after ";". Do it for all the future instances.

**Response:**

We have made corrections in the revised manuscript.

Line 30: Put a space between an in-text reference and the other text.

**Response:**

We have made corrections in the revised manuscript.

Line 31: Replace "region" with "regional".

**Response:**

We have made corrections in the revised manuscript.

Line 33: As per your stated reference, it's called "IGGDCB" where IGG stands for Institute of Geodesy and Geophysics. Please review.

**Response:**

It's called IGGDCB (IGG stands for Institute of Geodesy and Geophysics, Chinese Academy of Sciences)
We have made corrections in the revised manuscript.

Line 45-46: "It should be noted that some of the ….." Sentence needs to be rephrased to make it clearer.

**Response:**

We have revised it as following:

"It should be noted that these MGEX stations are selectively used in the DCB estimation of DLR and CAS, and the stations used by DLR and CAS are different. In other words, only the receiver DCB of stations used in DLR and CAS can be provided in their DCB products."

Line 49: "may be various" needs to be replaced with "may varies"

**Response:**

We have made corrections in the revised manuscript.

Line 52: "by Li et al. (2018)" instead of "Li (Li et al., 2018)."

**Response:**

We have made corrections in the revised manuscript.

Line 57: Point not clear. Please elaborate.

**Response:**

We have revised it as following:

"With the rapid development of MGEX, the BDS-3 measurements can be tracked by more MGEX stations."

Line 60: "The rest of the paper …" instead of " This rest of the paper …"

**Response:**

We have made corrections in the revised manuscript.

Line 65: "are presented" instead of " are shown."

**Response:**

We have made corrections in the revised manuscript.

Line 78: Put a full stop after receiver $r$ i.e. "$r$. The slant total electron content can be converted to …"

**Response:**

We have made corrections in the revised manuscript.

Line 81: Put a numerical value against "R" and "H". Refer to Schaer (1999).

**Response:**

We have added the numerical value (R=6371km and H=450km) in the revised manuscript.

Line 82: Get rid of "the" after "corresponding."

**Response:**

We have made corrections in the revised manuscript.

Line 94: Put the equation like this: m = m1 + …… + mr. It would have been better to use the subscripts here like m1, m2, mr.

**Response:**

We have made corrections in the revised manuscript.

Line 102: Put a space between 20° and cutoff.

**Response:**

We have made corrections in the revised manuscript.

Line 106-109: How has this been carried out? Existing software or new software written. Please add some information about it.

**Response:**

The MATLAB processing program is used in the paper, which is developed based on the M_DCB software (Jin et al., 2012)
We have added it in the revised manuscript.

Line 117: "And the corresponding …." Don't start a sentence with the conjunction 'and'. Remove all future instances.

**Response:**

We have made corrections in the revised manuscript.

Line 118: "epochs" instead of "epoch."

**Response:**

We have made corrections in the revised manuscript.

Line 127: What is the period/duration over which this mean is computed? It must be stated somewhere in the text.

**Response:**

The period is 30 days since we use 30 days of BDS observations to estimate the DCB. We have stated in the section 2.2.

Line 135: It should be "receiver DCB" instead of "receive DCB".

**Response:**

We have made corrections in the revised manuscript.

Figure 4 and Figure 5: Put labels to differentiate between CAS and DLR based estimation.

**Response:**

We have added the labels in the figures.

Figure 7: Have you tried comparing the trend of the estimated DCBs with the GIM you have used? After all, GIMs have certain accuracy. The ionosphere follows this pattern of being least active during the night to highly active during the day. So, this could influence the estimation process and one might be able to pick this trend by comparing the plotting the estimated DCBs against the ionospheric TEC derived from the GIM.

**Response:**

According to your suggestion, the trend of the estimated DCBs and the TEC is shown in the following figure. As shown in the figure, there is no significant correlation between the estimated DCBs and the TEC for the selected 9 stations within 4 days.

[Figure]

Figure. The receiver DCB (blue line) and VTEC of station (red line) for the selected 9 stations within 4 days

Comment# My experience of working with DCBs is that these are quite stable over time and especially in a controlled environment. One can see big fluctuations if there is some hardware problem in the receiver circuitry. In my opinion, most of these MGEX stations are in a relatively controlled environment, so the chances of all the receivers behaving in the same manner are quite low. It would be good to explore this area further.

**Response:**

Yes, the receiver DCB is relatively stable, while it is sensitive to the change of the hardware. Besides, the receiver DCB may varies within one day due to varying space environments and temperatures. The estimation of receiver DCB as constant every day may cause errors in ionospheric modelling, if the receiver DCB has significant intra-day fluctuations. It would have been better to analyze the intra-day variation of receiver DCB before the estimation of receiver DCB as constant over a day. Thus, the intra-day variation analysis of BDS receiver DCB with the additional BDS-3 observations is carried out in the study.

Please try plotting and comparing the DCBs for the stations which are utilising the same receiver type. Sept PolaRx5 could be good option with data available from 31 stations. I expect to see close results for these stations, no identical.

**Response:**

As your suggestion, we plotted the receiver DCB for the stations with the same receiver type, and the DCB products of CAS are also shown for the purpose of comparison. It's clear that the values of all receivers are not close. This indicates that the receiver DCB may be affected by many factors, and has no obvious relationship with receiver type.

[Figure]

Figure. The receiver DCB for the stations with the same receiver type (Sept PolaRx5) on January 11, 2019